

# Iflavirus increases its infectivity and physical stability in association with baculovirus

Agata K. Jakubowska[1,2], Rosa Murillo[3,6], Arkaitz Carballo[3,6], Trevor Williams[4], Jan W.M. van Lent[5], Primitivo Caballero[3,6] and Salvador Herrero[1,2]

[1] Department of Genetics, Universitat de València, Burjassot, Valencia, Spain
[2] Estructura de Recerca Interdisciplinar en Biotecnologia i Biomedicina (ERI-BIOTECMED), Universitat de València, Burjassot, Valencia, Spain
[3] Instituto de Agrobiotecnología, Universidad Pública de Navarra, Pamplona, Spain
[4] Instituto de Ecología AC, Xalapa, Mexico
[5] Laboratory of Virology, Dept. of Plant Sciences, Wageningen Agricultural University, Wageningen, Netherlands
[6] Departamento de Producción Agraria, Universidad Pública de Navarra, Pamplona, Navarra, Spain

Corresponding author
Salvador Herrero, sherrero@uv.es

## ABSTRACT

Virus transmission and the prevalence of infection depend on multiple factors, including the interaction with other viral pathogens infecting the same host. In this study, active replication of an iflavirus, *Spodoptera exigua iflavirus 1* (order *Picornavirales*) was observed in the offspring of insects that survived following inoculation with a pathogenic baculovirus, *Spodoptera exigua* multiple nucleopolyhedrovirus. Tracking the origin of the iflavirus suggested the association of this virus with the occlusion bodies of the baculovirus. Here we investigated the effect of this association on the stability and infectivity of both viruses. A reduction in baculovirus pathogenicity, without affecting its infectivity and productivity, was observed when associated with the iflavirus. In contrast, viral association increased the infectivity of the iflavirus and its resistance to ultraviolet radiation and high temperature, two of the main factors affecting virus stability in the field. In addition, electron microscopy analysis revealed the presence of particles resembling iflavirus virions inside the occlusion bodies of the baculovirus, suggesting the possible co-occlusion of both viruses. Results reported here are indicative of facultative phoresis of a virus and suggest that virus–virus interactions may be more common than currently recognized, and may be influential in the ecology of baculovirus and host populations and in consequence in the use of baculoviruses as biological insecticides.

## INTRODUCTION

Interactions between different viruses are often an inevitable consequence of multiple infection of a given host (*Berenyi et al., 2006*). Multiple infections are usually the result of consecutive infections by different viruses in the same host (known as super-infection), although simultaneous infections with two different viruses (known as co-infection)

have also been described (*Waner, 1994*; *Syller, 2012*; *Chen et al., 2004*). Virus–virus interactions occur at one of three levels: (i) direct interactions of viral genes or products, (ii) environmental interactions due to alterations of the host environment as consequence of the infection, and (iii) immunological interactions that result from the interaction with the host immune system (*DaPalma et al., 2010*). Independently of the type of interaction, the effect of the virus–virus interaction on viral fitness can generate a broad range of outcomes. For example, virus–virus interactions can have a positive effect on the fitness of both viruses (*Rizzetto, 2009*), or result in mutual exclusion (*Syller, 2012*). In contrast, non-autonomous viruses such as satellite viruses and virophages, occupy an intermediate position, as they depend on other viruses for replication and transmission, but both have negative effects on the production of their host viruses (*Krupovic & Cvirkaite-Krupovic, 2011*; *Wodarz, 2013*).

Baculoviruses are large DNA viruses that infect invertebrates, particularly insects of the order Lepidoptera (*Jehle et al., 2006*). Viruses of the genus *Alphabaculovirus*, i.e., nucle-opolyhedroviruses of Lepidoptera, are used worldwide as the basis for biological insecticides and as expression vectors for the production of recombinant proteins (*Szewczyk et al., 2006*; *Jarvis, 2009*). The baculovirus replication cycle involves two types of virions: occlusion derived virions (ODV) are responsible for the establishment of the primary infection in insect midgut cells, whereas budded virus (BV) is responsible for the systemic spread of infection within the insect. ODVs are embedded in a polyhedral occlusion body (OB), formed mainly by polyhedrin, that is responsible for protecting ODVs in the environment. During the infection process, the OBs dissolve in the gut of the insect host with the subsequent release of the ODVs that fuse to the midgut epithelial cell membranes and release nucleocapsids that are transported to the nucleus, starting the infective process (*Federici, 1997*).

Iflaviruses (family *Iflaviridae*, genus *Iflavirus*) are positive single-stranded RNA viruses that exclusively infect arthropods (*Van Oers, 2010*). Most iflaviruses produce inapparent sublethal infections in insect hosts, although some of these viruses can result in lethal infections in silkworms (*Aizawa & Kuruta, 1964*) and honeybees (*Ribiere, Olivier & Blanchard, 2010*). The number of described iflaviruses is relatively low, probably due to the lack of severe effects produced by most of these viruses (*Van Oers, 2008*). Only recently, the application of massive parallel sequencing methods has revealed, through the occurrence of expression sequence tags (ESTs) with homology to iflaviruses, the existence of new members of this family that were previously undetected (*Liu, Vijayendran & Bonning, 2011*; *Pascual et al., 2012*; *Oliveira et al., 2010*). The *Spodoptera exigua iflavirus 1* (SeIV1) was the first iflavirus described in *S. exigua.* SeIV1 has a genome of about 10 kb that codes for a 3222 amino acid polyprotein that, following proteinase processing, results in the structural and nonstructural viral peptides (*Millán-Leiva et al., 2012*). Despite its high infectivity, specificity and ability to replicate in *S. exigua* larvae, no clear effects on host fitness have been detected in insects infected with this virus (*Millán-Leiva et al., 2012*; *Jakubowska et al., 2014*).

In the present study we report that caterpillars (larvae) of the beet armyworm, *S. exigua*, the offspring of insects that survived following inoculation with an isolate of *Spodoptera exigua multiple nucleopolyhedrovirus* (SeMNPV), had acquired a persistent sublethal infection with SeIV1. This serendipitous result was obtained during a microarray study

on the expression of insect and viral genes in insects that had survived inoculation with baculovirus. Contrary to expectations, we did not detect persistent baculovirus infection, but instead observed high expression of iflavirus genes in these larvae. The source of the iflavirus infection was traced back to the SeMNPV OB inoculum. In the present study we present evidence that this association between the viruses results in increased environmental persistence and transmission opportunities for the iflavirus and a reduction in the pathogenicity of the baculovirus. Subsequent studies revealed that mixed infections may result from co-infection with particles of both viruses physically associated with each other.

## METHODS

### Insects and viruses

A virus-free *S. exigua* colony (SUI) was obtained from Andermatt Biocontrol AG (Grossdietwil, Switzerland) and reared in continuous culture at a constant temperature (25 ± 1 °C), relative humidity (RH; 50% ± 5%), and photoperiod (16-h/8-h light-dark cycle) on artificial diet (*Elvira et al., 2010*) in the insectary facilities of the Universidad Pública de Navarra (UPNa, Pamplona, Spain). Insects from the UPNa colony were used to start a sister colony reared at Universitat de Valencia (UV, Valencia, Spain) using similar conditions. Insects were routinely tested for the presence of SeIV1 and SeMNPV, and confirmed to be virus-free ahead and during each experiment.

SeIV1 was detected and isolated from the laboratory colony of *S. exigua* (*Millán-Leiva et al., 2012*). The SeMNPV isolate used for the establishment of covert infections was originally isolated from spontaneous infections observed in the laboratory-reared offspring of field-caught *S. exigua* females (*Cabodevilla et al., 2011*). For this study, a fresh stock of SeMNPV OBs was amplified *in vivo* by droplet-feeding *S. exigua* fourth instar larvae. OBs were purified from cadavers by washing in 0.01% SDS twice, once in double-distilled water and finally diluted in sterile double-distilled water. OB suspension was stored at −20 °C until required. OBs were quantified by counting in triplicate with a Neubauer chamber.

A SeIV1-free isolate of SeMNPV was obtained after PCR screening of the UPNa baculovirus collection. The Spanish isolate SeMNPV-SP2 was found to be negative for the presence of SeIV1 and was selected for subsequent studies. This isolate was originally obtained from a group of cadavers collected during a baculovirus epizootic in greenhouse crops in southern Spain (*Caballero et al., 1992*; *Muñoz et al., 1999*). OBs had been maintained at −20 °C since 1996.

OBs from the SeMNPV-SP2 isolate associated with SeIV1 (SeMNPV-SeIV1) and SeIV-free OBs (SeMNPV) were produced for biological comparison. For this, groups of 25 newly molted fourth instars were allowed to drink from a suspension of $10^4$ OBs/ml in a solution of 10% (w/v) sucrose and 0.001% (w/v) food dye. A second batch of 25 larvae was fed with 20 μl of an identical OB concentration mixed with 80 μl of $1.34 \times 10^{-1}$ ng/μl SeIV1 particles ($\approx 1.46 \times 10^7$ SeIV1 genomes/μl). Larvae were reared individually until death or pupation. OBs were purified from cadavers and tested for SeIV1 by RT-qPCR as described below. The SeIV1 load present in OBs produced by this method was estimated at 18.9 ± 2.3 SeIV1 genomes per OB.

## Infection with baculovirus

To infect *S. exigua* individuals with SeMNPV, larvae from a virus-free laboratory colony were challenged *per os* with an estimated 50% lethal concentration of OBs (*Cabodevilla et al., 2011*). Briefly, a batch of 30 pre-molt *S. exigua* third instars were starved overnight, allowed to molt to the fourth instar and then allowed to drink an OB suspension containing $9 \times 10^3$ OBs/ml during a 10-min period (*Hughes & Wood, 1981*). Control insects (VF) consumed droplets that did not contain OBs. Inoculated larvae were individually placed in 25-ml plastic cups perforated for ventilation and provided with artificial diet. OB-challenged (VT) larvae that did not succumb to polyhedrosis disease were reared through to pupation at 25 ± 1 °C and 50% ± 5% RH. According to previous experiments (*Cabodevilla et al., 2011*), these larvae were expected to carry a persistent baculovirus infection as the experiment was originally planned to examine the expression of baculovirus and insect genes in persistently infected larvae. We were, however, unable to confirm establishment of the persistent infection as originally planned, which led us to investigate the relationship between SeIV1 and SeMNPV. Pupae were sexed and, once adults emerged, one male–female pair of adults was placed in a paper bag, and allowed to mate and oviposit to obtain the subsequent generation (F1). An identical procedure was applied to mock-inoculated insects that were defined as control samples. Egg masses were collected and placed in 300-ml plastic containers provided with artificial diet until larvae reached the second instar. From these containers 25 larvae were individualized and reared through to the fifth instar as described above. All procedures were performed in triplicate. Four F1 fifth-instar larvae per replicate were randomly selected and pooled for RNA extraction and microarray analysis. For each sample (VT and VF), three different pools were obtained and used for subsequent gene expression analysis.

## Microarray design, hybridization and analysis

A 44K Agilent oligonucleotide microarray was designed to study different aspects of the interaction of *S. exigua* larvae with viral (iflavirus and baculovirus) and bacterial pathogens (*Jakubowska, Vogel & Herrero, 2013*; *Bel et al., 2013*). In that sense, the array mainly comprised probes representing unigenes from the *S. exigua* transcriptome (*Pascual et al., 2012*; *Jakubowska, Vogel & Herrero, 2013*), but also included 60-mer non-overlapping tiling probes covering both strands of the SeIV1 and SeMNPV genomes. In total, the microarray contained 167 probes covering the positive strand of SeIV1 and 166 probes covering the negative strand. The SeMNPV genome was represented by 2,260 tiling probes covering both virus strands. The microarray was also used to determine the expression levels of 139 open reading frames (ORFs) predicted for the SeMNPV genome (*Ijkel et al., 1999*). Two different 60-mer probes were included for each of the predicted ORFs. The probes were designed using the eArray application from Agilent.

Synchronized F1-larvae from the VT and VF groups were collected and total RNA was extracted using RNAzol reagent (Molecular Research Center, Inc., Cincinnati, OH), according to the manufacturer's protocol. To further purify RNA, an RNAeasy Kit (Qiagen, Hilden) was used following the protocol provided by the manufacturer. The quality of RNA was assessed by Agilent 2100 Bioanalyzer using the EukaryoteTotal RNA Nano protocol.

Agilent One-Color Spike-in Mix was added and 600 ng of total RNA was used for cRNA (complimentary RNA) synthesis. The obtained cRNA (1.65 μg) was fluorescently labeled with cyanine-3-CTP, fragmented and hybridized to *S. exigua* microarray slides following the One-Color Microarray-Based Gene Expression Analysis (Quick-Amp labelling) protocol. Microarrays were scanned using G2505B Agilent scanner and data were extracted using Agilent Feature Extraction 9.5.1 software. Spike-in transcripts are a mix of unique 55-mer probes that specifically anneal to complementary control probes on the Agilent's microarrays and were used for linear normalization performed by the Agilent Extraction 9.5.1 software. Before data analysis, hybridization quality control reports were verified as correct. RNA labelling and hybridization, as well as array scanning and data extraction were performed by the Microarray Analysis Service of Principe Felipe Research Centre (CIPF, Valencia, Spain) following standard protocols.

Data analysis was performed using Babelomics 4.3 software (http://babelomics.bioinfo.cipf.es/) (*Medina et al., 2010*). First, between-arrays normalization was performed using the quartile normalization method in Babelomics (*Bolstad et al., 2003*). Normalized arrays of the VT samples were compared to VF controls and expressed as a fold-change in gene expression or abundance. For those probes having low signal levels for one of the samples (as indicative of absence in one of the samples), changes in gene expression were estimated by comparison with the overall background intensity.

## SeIV1 detection by RT-PCR

RT-qPCR was used to detect SeIV1 genomic RNA in the purified preparations of SeMNPV and SeIV1. SeMNPV occlusion bodies (OBs) as well as SeIV were purified on discontinuous sucrose gradients as described below. RNA was extracted from the samples using Tripure reagent (Roche), according to the manufacturer's protocol. RNase-free glycogen (5 μg/μl) was added as a carrier during the RNA precipitation step. Purified RNA was used for cDNA synthesis using PrimeScript RT reagent kit from Takara Bio Inc. (Otsu Shiga, Japan) following the manufacturer's protocol. RT-qPCR was carried out in a StepOnePlus Real-Time PCR System (Applied Biosystems, Foster City, CA). All reactions were performed using HOT FIREPOL EvaGreen qPCR mix Plus (ROX) from Solis BioDyne (Tartu, Estonia), in a total reaction volume of 25 μl. Forward and reverse primers, designed using Primer Express software (Applied Biosystems, Foster City, CA), were added to a final concentration of 0.3 μM. These specific primers were designed to amplify a 97-bp fragment in the RNA-dependent RNA polymerase (RdRp) region from 9743-9840 nt on the genome (Forward: 5′-TGTGAAGTTAGACACGCATGGAA-3′ and Reverse: 5′-CGACTTGTGCTACTCTCTTCATCAA-3′). For relative quantification of virus genomes, Ct values from the RT-qPCR were compared to the standard curves obtained for known number of copies of virus genome fragment cloned in the pGEMTeasy vector. A fragment of SeIV1 genome was cloned into pGEMTeasy vector and the standard curve prepared from the serial dilutions of known copies of the vector DNA.

Semiquantitative RT-PCR was used to detect the negative RNA strand of SeIV1 in larvae as well as in SeMNPV OBs. Tagged primer was used for the specific synthesis of cDNA due to the occurrence of self-priming, often observed for RNA viruses. RNA was extracted

as described above. For this, 0.5 μg of RNA were used for cDNA synthesis using tagged specific primer (5′ -ggatgcaggctacgtgaagatacgGTGTCAACAACAGACCCTAGCG-3′, tag in lowercase, SeIV1 specific sequence in uppercase). cDNA synthesis was performed using PrimeScript RT reagent kit from Takara Bio Inc (Otsu Shiga, Japan) following the manufacturer's protocol, at 42 °C for 30 min. 2 μl were used for subsequent PCR reaction using the following primers: forward 5′ -ggatgcaggctacgtgaagatacg-3′ and reverse 5′ -gcagccatgttcaacctc-3′ , and the following conditions: 94 °C for 5 min, annealing at 55 °C and elongation at 72 °C for 30 cycles. The resulting PCR product had a size of 1,495 bp.

## Virus purification by gradient centrifugation

For electron microscopy, OBs from SeMNPV and SeIV1-associated SeMNPV samples were additionally purified through sucrose gradients (*King & Possee, 1992*). Briefly, 3 ml of $\sim 10^8$ OB/ml suspensions were loaded onto a 30–60% (w/w) continuous sucrose gradient and then centrifuged at $40,000 \times$ g for 1 h at 4 °C. The OB band was harvested by puncturing the tube with a needle and collecting the sucrose fraction in a syringe. OBs in sucrose were diluted in 2 vol. of $1\times$ TE buffer and centrifuged at $40,000 \times$ g for 1 h. OBs collected in the pellet were suspended in sterile double-distilled water.

SeIV1 particles were purified from *S. exigua* larvae as follows. Approximately one hundred fourth and fifth instar larvae were freeze-killed and lyophilized. The lyophilized larvae were homogenized in 0.01 M potassium phosphate pH 7.4 containing 0.45% diethyldethiocarbamic acid (DIECA) and 0.2% $\beta$-mercaptoethanol (2–5 ml of buffer per gram of larvae). The homogenate was then sonicated for 20 s and subsequently filtered through a double layer of cheesecloth and then centrifuged for 15 min at $5,000 \times$ g to remove large debris. The supernatant was centrifuged for 2.5 h at $82,000 \times$ g (Beckman centrifuge, S28 rotor). The pellet was resuspended in 2 ml TAE (40 mM Tris, 20 mM acetic acid, 1 mM EDTA) buffer at pH 7.3 and left overnight at 4 °C. The suspension was then applied onto 20% sucrose in TAE buffer and centrifuged 2.5 h at $100,000 \times$ g at 4 °C. The pellet was again resuspended in a small volume of TAE buffer and left overnight at 4 °C. Next, 10–40% discontinuous sucrose gradients were prepared and the sample was centrifuged for 2.5 h at $100,000 \times$ g. SeIV1 purified particles were collected from the white virus fraction.

## Determination of SeIV1 infectivity in larvae

In order to compare the ability of SeIV1 alone and associated with SeMNPV OBs to enter and replicate in host cells, virus-free *S. exigua* fourth instar larvae were starved overnight and then orally inoculated by allowing them to drink from an OB suspension of SeMNPV-SeIV1 containing $1 \times 10^6$ OBs/ml by the droplet-feeding method (*Hughes & Wood, 1981*). Another batch of larvae was orally inoculated with the same amount of SeIV1 particles that were estimated to be present in the SeMNPV-SeIV1 OBs (estimated by RT-qPCR). Both SeMNPV-SeIV1 OBs and SeIV1 particles used in the experiment originated from two separate preparations. At 72 hpi larval midguts were dissected and SeIV1 loads measured by RT-qPCR as described above. Ten larvae were infected with each virus preparation and ten larvae served as controls, to confirm that the SUI colony was virus-free, and that the insects were not contaminated during the experiment. Larvae were

processed individually. Ct values of SeIV1 were normalized to Cts for the ATP synthase reference gene (*Herrero et al., 2005*), and $-\Delta$ Ct values compared between larvae infected with SeIV1 alone or associated with OBs. The resulting $-\Delta$ Ct values were compared by Mann–Whitney test, due to the presence of outliers.

## Determination of dose-mortality response

The pathogenicity of SeMNPV OBs and SeMNPV-SeIV1 OBs was determined by the droplet-feeding method (*Hughes & Wood, 1981*). Briefly, groups of 30 newly molted second instars were starved overnight and orally inoculated with one of the following OB concentrations: $2.54 \times 10^5$, $8.18 \times 10^4$, $2.72 \times 10^4$, $9.09 \times 10^3$ and $3.03 \times 10^3$ OBs/ml. This range of concentrations was estimated to kill between 95% and 5% of the experimental insects. Larvae that ingested droplets within 10 min were individually transferred to a 24-well tissue culture plate. A cohort of 24 larvae was allowed to drink from an OB-free suspension as controls. Larvae were reared on a semisynthetic diet at 25 $\pm$ 2 °C and mortality was recorded daily for 7 days post-inoculation. The entire bioassay was performed three times. Data were subjected to Probit regression analysis using the Polo-PC program (Le Ora Software, 2002). Lethal concentration ($LC_{50}$) values and relative potencies were estimated when a parallelism test confirmed that the regressions for each treatment could be fitted with a common slope (*Robertson & Preisler, 1992*).

## Determination of OB production and virulence

Groups of 30 fourth instar larvae were starved overnight and allowed to drink for 10 min from an OB suspension containing $5 \times 10^7$ OBs/ml of either SeMNPV/SeIV+ or SeMNPV/SeIV1-. Individual weight measurements were first taken immediately before inoculation. Inoculated larvae were individualized in 24-well plates containing diet and checked daily for virus-induced mortality. Larvae were monitored every eight hours for mortality and weighed daily during a six day post-inoculation period. Virus-killed larvae were frozen at $-20$ °C to avoid liquefaction. Cadavers were individually homogenized in 1 ml sterile distilled water. Each homogenate was filtered through cheesecloth to remove debris and the resulting suspension was counted in triplicate in a Neubauer chamber at $400\times$ magnification using a phase contrast microscope. The experiment was performed six times. OB production and weight gain data were not normally distributed, and were compared by Kruskall–Wallis or Mann–Whitney test (SPSS statistics V.21, 2012).

## Effect of UV and temperature treatment on the stability of SeIV1

The effect of environmental factors such as UV radiation and high temperature on the stability of SeIV1, alone or in association with OBs, was indirectly estimated by genome integrity measured by RT-qPCR. To this aim, two preparations of SeIV1 (SeMNPV OBs containing SeIV1 or SeIV1 alone) were purified by sucrose gradient centrifugation and then exposed to different intensities of UV-C radiation or temperature for different periods of time. The amounts of SeIV1 that remained viable for amplification were estimated by RT-qPCR.

For the UV irradiation treatment, each of the SeIV1 preparations was exposed to 0, 3, 6, 9 and 12 J/cm$^2$ UV-C light using a crosslinker CL-1 (Herolab), at a wavelength of 254 nm.

Samples of 200 µl of each preparation were placed in 24-well plates, to achieve a suspension depth of ~1 mm, and were then exposed to continuous UV-C light 30 cm below the lamp, which allowed an exact dose to be administered to each sample. At each sample time, 300 µl of Tripure was immediately added to 150 µl of the sample, and the samples were frozen for further RNA purification. RNA was extracted and cDNA synthesized as described above. RT-qPCR was used to detect the presence and to estimate the number of SeIV1 genomes in the samples. The experiment was performed twice and the data were analyzed by two-way Anova.

For the heat treatment, each SeIV1 preparation was incubated at 72 °C for 0, 30, 60, 360 and 1440 min in an Eppendorf thermomixer. As before, at each sample time 300 µl of Tripure was immediately added to 150 µl of the sample, and then frozen until RNA purification. RNA was extracted and cDNA synthesized as described above. RT-qPCR was used to detect the presence and estimate the SeIV1 load in each sample. The experiment was repeated twice and the data were analyzed by two-way Anova. Starting quantities of SeIV1 genomes in all samples were normalized to 100%, and the decrease in estimated quantities at each sample time calculated accordingly.

### Electron microscopy

Scanning electron microscopy (SEM) was used to examine the presence of SeIV1 particles on the surface of OBs. For this, OBs in suspension were fixed overnight by mixing with an equal volume of fixative (4% formaldehyde and 1% glutaraldehyde in 0.1M phosphate buffer, pH 7.4) and then washed twice with 0.1M phosphate buffer. Samples were then partially dehydrated with 70% ethanol, dried, placed on aluminum mounts using carbon tags, sputter-coated with gold-palladium and photographed at magnifications of 6,000× and 25,000× using a scanning electron microscope (Philips SEM 550).

Transmission electron microscopy (TEM) was used to examine the polyhedrin matrix within OBs. For this, OBs in suspension were fixed for 2 h at 4 °C with 1.5% glutaraldehyde. The samples were then concentrated in 0.4% agar and washed with phosphate buffer (0.2 M, pH 7.3). Samples were post-fixed with 2% osmium tetroxide ($OsO_4$) for 2 h, dehydrated and stained for 1 h with 2% uranyl acetate. The samples were then embedded in epoxy resin and polymerized for 48 h at 60 °C. After polymerization, samples were sectioned using an ultramicrotome (Leica UC6), transferred to TEM grids and stained with lead acetate. The resulting grids were observed under an electron transmission microscope of 100 kV (JEOL JEM 1010). Different fields of each sample were photographed at a magnification of 40,000× and visualized with image acquisition software.

## RESULTS

### Detection of iflavirus expression in *S. exigua* larvae treated with baculovirus

Gene expression comparison between *S. exigua* larvae derived from insects previously inoculated with SeMNPV (VT), or virus-free larvae (VF), was performed using a custom-made DNA-microarray containing *S. exigua* unigene probes; about 3,000 probes covering the complete genomes of SeMNPV and SeIV1 (tiling probes), and the predicted ORFs

from SeMNPV. Microarray comparison showed high differential expression of the probes representing the positive (average log2 value of 11.1) as well as the intermediate (negative) strand (average log2 values of 7.2) of the SeIV1 genome (Fig. 1A), indicating the presence and active replication of SeIV1 in the offspring of insects that survived oral inoculation with SeMNPV OBs. In contrast, the presence of SeMNPV transcripts was not detected (ratio VT/VF equal to 1) in the VT insects (Figs. 1B and 1C) which also indicates that the iflavirus was capable of autonomous replication in the absence of baculovirus transcription.

## Influence of the virus association on iflavirus infectivity and baculovirus pathogenicity and virulence

Semi-quantitative PCR as well as reverse transcription quantitative PCR (RT-qPCR) revealed the presence of SeIV1 genomes in the SeMNPV OB preparation that had been purified after several centrifugation steps prior to being used to inoculate VT insects. Both positive and negative strand of the virus were detected, nevertheless the negative strand was present only in trace amounts when compared with its abundance in the larvae. This finding was indicative of a possible association between both viruses. We decided to explore the influence of such association on the insect–virus relationship of both types of virus. To determine whether iflavirus association with baculovirus favors iflavirus infectivity, the ability to establish an infection in host insects was estimated using iflavirus inoculum alone or associated with SeMNPV OBs. For this, SeIV1-free insects were orally inoculated with a preparation of SeMNPV OBs containing SeIV1 particles (quantified by RT-qPCR). In a side-by-side approach, a second batch of virus-free insects was inoculated with the same amount of SeIV1 particles, quantified by RT-qPCR method described above, but in the absence of SeMNPV OBs. Three days after inoculation, insects were dissected and the abundance of SeIV1 genomes in the midgut of the larvae, as an estimation of the SeIV1 ability to establish a viral infection, was determined by RT-qPCR. Results revealed that the iflavirus was present in all inoculated insects, but a 2.8-fold difference in SeIV1 load, indicated by a 1.5-fold difference in qPCR Ct values, was detected in insects inoculated with iflavirus associated with OBs than when inoculated alone (Fig. 2A). Control insects were confirmed to be negative for both viruses.

We also determined the effect of the viral association on the virulence and pathogenicity of SeMNPV OBs. In contrast to the results for iflavirus, the pathogenicity of SeMNPV OBs, estimated by peroral bioassay, was reduced by ∼40% when iflavirus was associated with OBs compared to equal OB inoculum free from iflavirus (Fig. 2B). However, the co-transmission of iflavirus and SeMNPV did not significantly affect the mean speed of kill of SeMNPV that varied between 98 and 102 h post-inoculation (Fig. 2C). Similarly, the growth of infected host insects (Fig. 2D-left), or the total OB production (Fig. 2D-right), in insects inoculated by baculovirus OBs with iflavirus, did not differ significantly from that of insects inoculated with iflavirus-free OBs.

## Influence of the virus association on iflavirus persistence in the environment

Occlusion of baculovirus ODVs greatly improves the persistence of these virions in the environment following death of the insect host (*Granados & Federici, 1987*). If SeIV1 is

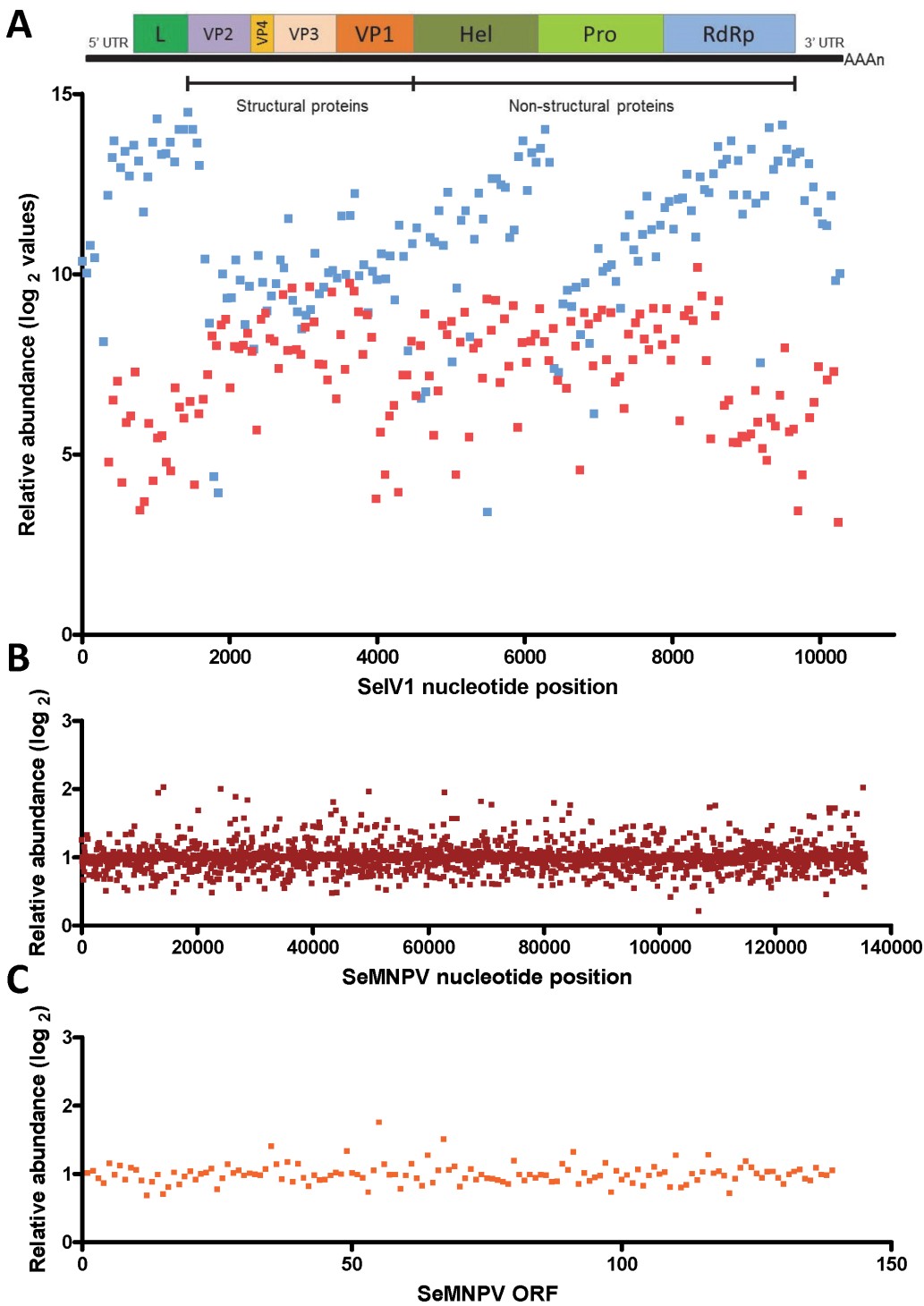

**Figure 1  Abundance of viral sequences in the progeny of insects sublethally infected with SeMNPV, (virus treated, VT), measured by tiling array of the viral genomes.** Values are the ratio of normalized intensity between VT and virus free (VF) samples for each of the studied probes. (A) Abundance of SeIV1 sequences and their position along the genome. Genome structure is represented at the top of the panel and aligned according to the position of the corresponding 60-mer probe. (continued on next page…)

**Figure 1 (…continued)**
Each spot represents the relative abundance of sequences hybridizing to each probe. Abundance is reported as log2 values, which means that a difference of 10 in log2 values corresponds to a 1024-fold difference. Blue and red spots represent the abundance of sequences from the positive and the intermediate (negative) strand of the SeIV1, respectively. Electrophoresis panel reflects the detection of the negative strand of SeIV1 in larvae (L) and baculovirus OBs (OB) by semiquantitative RT-PCR. (B) Abundance of SeMNPV transcripts and their position along the genome. Each spot represents the relative abundance of sequences that hybridized to each 60-mer probe. (C) Expression of SeMNPV ORFs (GenBank, NC_002169.1) in VT insects. Each ORF value represents the average of at least two probes.

detected in association with the baculovirus occlusion body, it may be expected that such an association could contribute to increasing its stability in the environment. In agreement with such a hypothesis, experiments involving exposure to an ultraviolet light source (UV-C, 254 nm wavelength) or high temperature (72 °C) revealed that SeIV1 particles associated with OBs maintained physical stability significantly better than naked particles. Following exposure to 3–12 J/cm$^2$ of UV-C radiation, the stability of occluded iflavirus particles, measured as the relative viral load, was approximately two orders of magnitude greater at each time point than that of naked particles (Fig. 2E). Similarly, exposure of iflavirus particles to high temperature resulted in ∼100-fold greater stability of OB-associated iflaviruses, following 1 or 6 h exposure, compared to naked particles (Fig. 2F).

### Detection of the iflavirus in baculovirus OBs

RT-qPCR revealed the presence of SeIV1 genomes in the SeMNPV OB preparation that had been purified by several centrifugation steps prior to being used to inoculate larvae. Moreover, the iflavirus particles present in the preparation were able to establish a persistent infection. This finding suggested that a physical association may exist between both viruses. The physical association between the viruses is likely to involve the localization of iflavirus particles inside or outside of OBs. To determine this, purified OBs from insects that died of SeMNPV infection in the presence of iflavirus (SeIV1+), or the absence of iflavirus (SeIV1−), were observed by scanning electron microscopy (SEM) (Figs. 3A–3H). No differences were observed in the external appearance of each type of OB. In contrast, transmission electron microscopy (TEM) of the OBs revealed the presence of ODVs of baculovirus, each comprising 1–4 nucleocapsids (Figs. 4A–4C), as well as dark spots dispersed in the matrix resembling in size and form (Figs. 4D and 4E), the icosahedral particles of iflavirus embedded in the polyhedrin matrix (Fig. 4F). To confirm this, the TEM procedure was repeated in an independent laboratory (Laboratory of Virology of Wageningen University, the Netherlands) to exclude the possibility of generating artifacts during sample preparation. Similarly, samples processed in the Netherlands laboratory showed the presence of dark spots resembling iflavirus particles in a certain number of OBs (Figs. 4G and 4H).

## DISCUSSION

Traces of sequences from the SeIV1 genome were first identified in the *S. exigua* larval transcriptome (Pascual et al., 2012). Additional studies revealed that SeIV1 can replicate, disperse through larval feces, and be horizontally transmitted with a very high efficiency

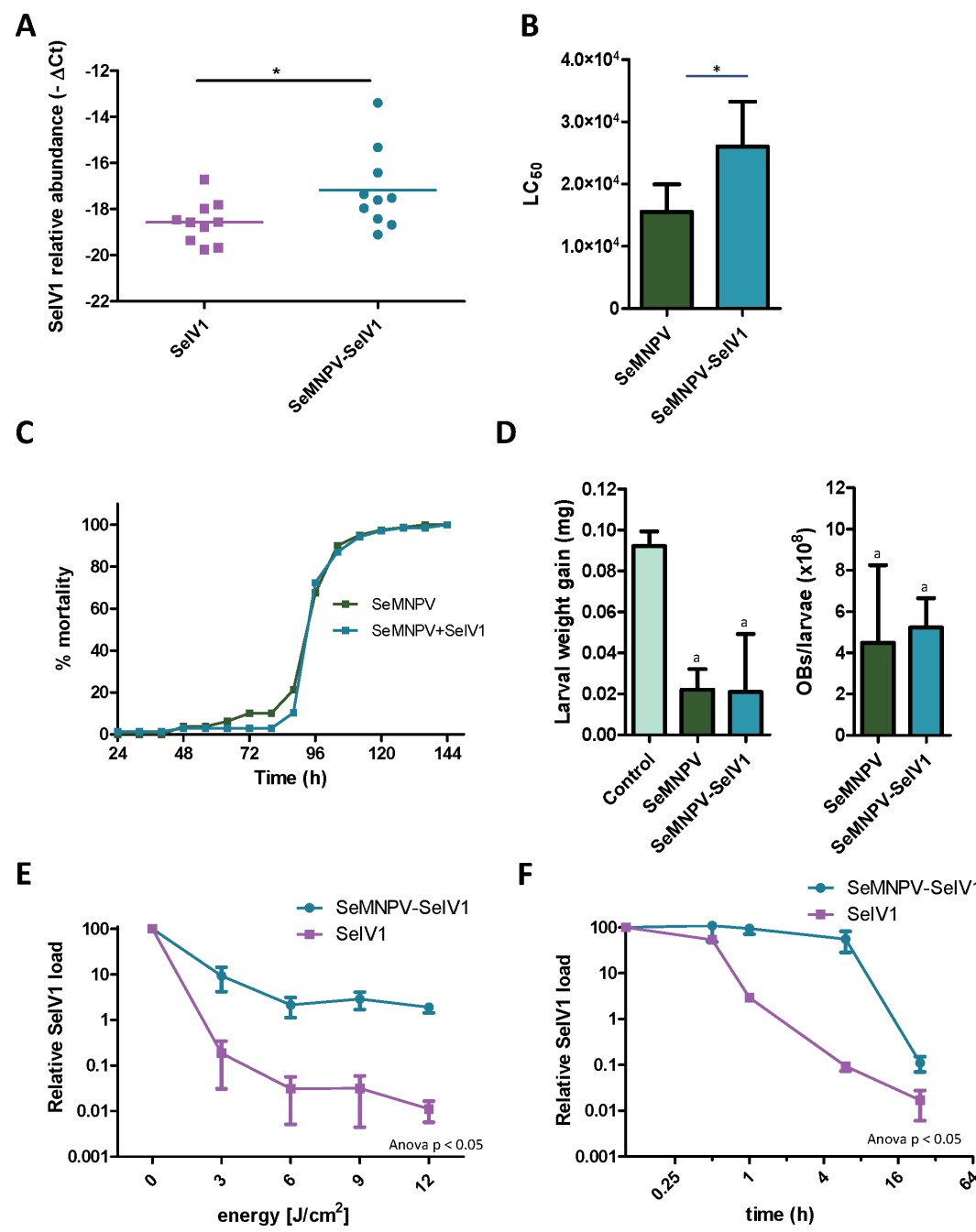

**Figure 2  Stability, replication and host growth effects of SeIV1 and SeMNPV in association or alone.**
(A) SeIV1 infection alone or associated with SeMNPV OBs after oral inoculation of *S. exigua* larvae. Asterisk indicates significant difference ($P < 0.05$). (B) Pathogenicity of SeMNPV OBs alone and associated with iflavirus expressed in $LC_{50}$ (OBs/ml). Asterisk indicates significant difference ($P < 0.05$). (C) Virulence of SeMNPV OBs alone and associated with iflavirus expressed as mean time to death (MTD). MTD values were estimated by Weibull survival analysis (*Crawley, 1993*). Curves did not differ significantly (*t*-test, $P = 0.08$). (D) Larval weight gain after oral inoculation with SeMNPV OBs alone or OBs in association with SeIV1 particles and SeMNPV ODVs (D-left). Production of OBs in *S. exigua* larvae infected with SeMNPV OBs alone or in association with SeIV1 particles (D-right). Means with the same letter are not significantly different ($P > 0.05$). Relative SeIV1 load alone or associated with OBs when exposed to different doses of ultraviolet light (E) or periods of heating at 72 °C (F).
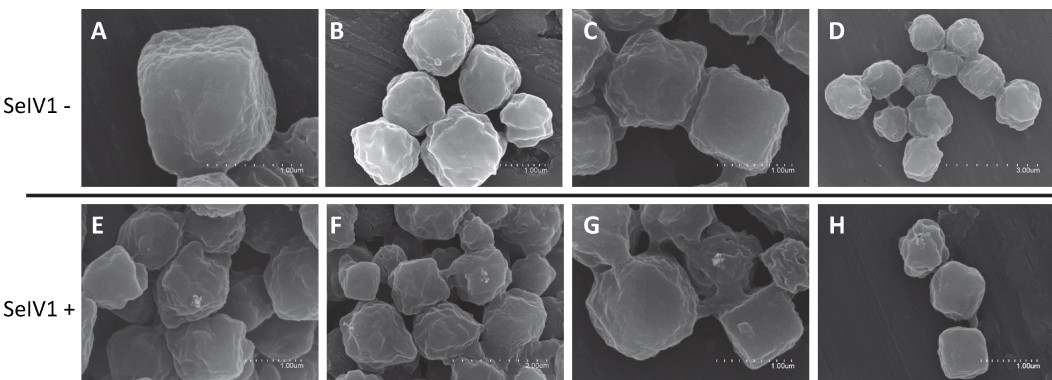

**Figure 3** **SEM images of purified SeMNPV OBs.** Representative images from SeIV1-free OBs (A–D) and SeIV1-containing OBs (E–H).

but without clear pathological effects on the host (*Millán-Leiva et al., 2012*). Horizontal transmission in the field mainly depends on the prevalence of infected insects (as a source of inoculum), the density of susceptible hosts and on viral resistance to abiotic factors that could negatively impact the persistence of virus particles in the environment. In the present study we obtained clear evidence that iflavirus SeIV1 was present in the baculovirus OB preparation, even after extensive purification of the OBs by a series of sucrose gradient centrifugations. Moreover, this association resulted in the establishment of persistent iflavirus infections in insects that consumed baculovirus OBs.

In certain situations, simultaneous infection with different species of baculovirus or baculovirus with other invertebrate viruses can increase the effectiveness of these pathogens as agents of biological control (*Washburn et al., 2000*; *Guo et al., 2007*). However, very few studies have focused on determining the effect of mixed infections on indicators of viral fitness. For example, mixed infection of the lepidopteran *Adoxophyes honmai* with a nucleopolyhedrovirus (AdhoNPV) and an entomopoxvirus (dsDNA virus) resulted in a reduction in the fitness of both viruses (*Ishii et al., 2002*). In contrast, early studies on the interaction of an alphabaculovirus (AcMNPV) with an unclassified RNA virus of *Trichoplusia ni* revealed that RNA virus-infected larvae had reduced growth compared to healthy insects, but with little or no significant effects on the pathogenicity or speed of kill of alphabaculovirus OBs against host larvae (*Vail, Morris & Collier, 1983*). Another interaction has been recently described between viruses infecting *Helicoverpa armigera* (*Xu et al., 2014*). A clear negative interaction between *H. armigera* densovirus 1 (HaDNV1) and a baculovirus (HaSNPV), was observed in wild populations of this pest. Laboratory assays revealed that HaDNV1-infected insects were significantly more resistant to HaSNPV infection than non-infected insects, suggesting a mutualistic relationship between the host insect and the HaDNV1 virus (*Xu et al., 2014*). Similarly, a recent study on field-collected *S. exigua* insects detected simultaneous covert infections with SeMNPV and iflaviruses in around 10% of the captured adults, while about 40% of the insects were covertly infected with SeMNPV alone (*Virto et al., 2014*). These findings may suggest that a similar mutualistic interaction between *S. exigua* and the iflavirus exists in wild populations of

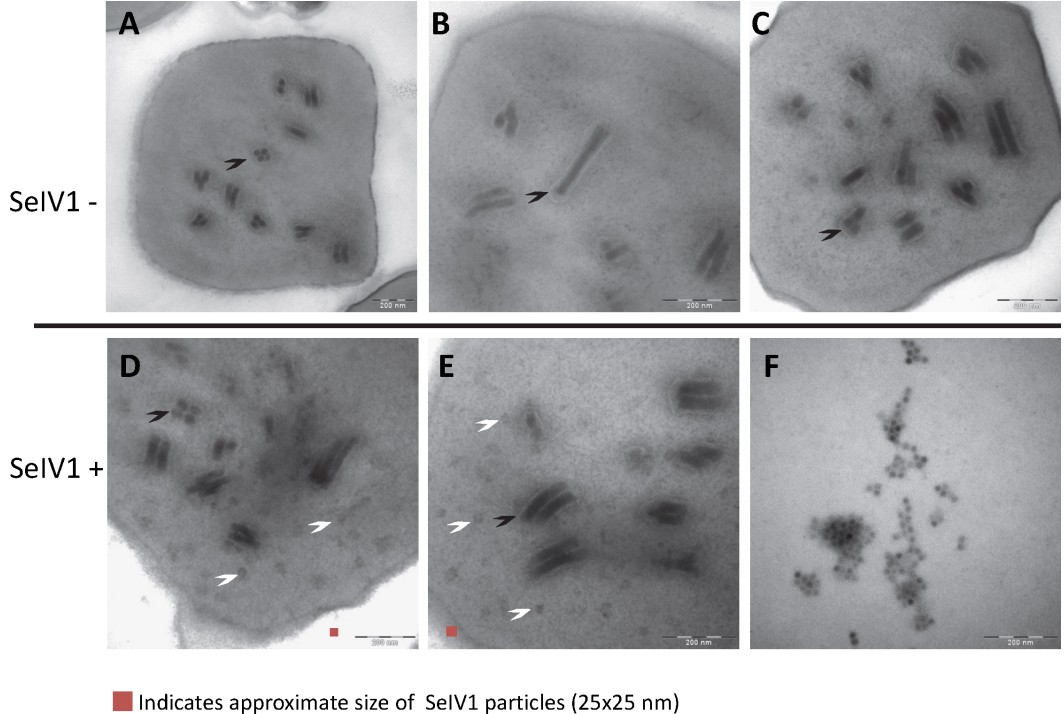

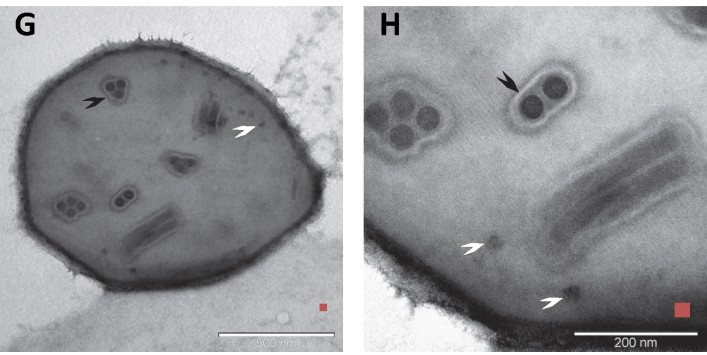

■ Indicates approximate size of SeIV1 particles (25x25 nm)

**Figure 4  TEM images of purified OBs from SeMNPV and SeIV1.** Representative images from SeIV1-free SeMNPV (A–C), SeIV1-containing SeMNPV (D, E), and purified SeIV1 particles (F). TEM was performed at the University of Valencia, Spain. In order to exclude the possibility of a methodological artifact, TEM of SeIV1-containing SeMNPV was also performed at Wageningen University, The Netherlands (G, H). Some ODVs (black arrowhead) and SeIV1-like particles (white arrowhead) are indicated. For size comparison, a red square of 25 nm per side is shown in each figure.

this pest. In the present study we found that the presence of the iflavirus was slightly detrimental to the baculovirus, as iflavirus-contaminated OBs were less pathogenic in healthy insects, compared to OBs that contained SeMNPV ODVs alone. In contrast, by physical association with the baculovirus, SeIV1 particles appear capable of extending their survival in the environment and are more likely to infect the host when consumed by a susceptible insect. The last result has to be taken with caution, since the method of quantification of the IV from the two different matrices (purified IV and IV present in OBs) may be biased.

Results reported here demonstrate a possible facultative phoresis of one virus by another and suggest that the interaction of these viruses is relevant for the transmission and replication of both viruses. Iflaviruses have generally received little attention from invertebrate pathologists due to their low pathogenicity and tendency to produce inapparent sublethal infections (*Millán-Leiva et al., 2012*; *Chen, Becnel & Valles, 2012*). Co-infection of iflavirus with baculoviruses in field conditions has been reported recently (*Virto et al., 2014*). However, as the number of novel RNA viruses, including iflaviruses, increases rapidly through the use of massively parallel sequencing methods (*Liu, Vijayendran & Bonning, 2011*; *Jakubowska et al., 2015*; *Carrillo-Tripp et al., 2014*; *Geng et al., 2014*), the discovery of novel interactions in natural virus populations is likely to grow accordingly. Our results suggest that virus–virus interactions may be more common than currently recognized, and may be influential in the ecology of baculovirus and host populations. In this respect attention has heavily focused on virus-pathogen interactions in honeybees due to growing concerns over colony collapse disorder (*Cornman et al., 2012*). In another case of co-infecting microorganisms, persistent infection with *Wolbachia* has been reported to protect against infection by RNA viruses in dipterans (*Glaser & Meola, 2010*), but increases mortality due to baculovirus infection in a lepidopteran (*Graham et al., 2012*).

In this study, although using similar conditions and viral concentrations as employed in previous experiments for the generation of persistent infections (*Cabodevilla et al., 2011*), we could not detect baculovirus transcription. It is possible that the presence of SeIV1 in association with SeMNPV negatively affected the establishment of persistent infection by the baculovirus. The presence of iflavirus associated with the baculovirus is therefore likely to affect the dynamics of baculovirus transmission in natural *S. exigua* populations and could also affect the insecticidal properties of baculoviruses used as biological insecticides. Decreased pathogenicity as a result of the presence of iflavirus in association with OBs might also reduce the establishment of persistent baculovirus infections since fewer individuals are likely to become infected and viral dissemination will be limited. Moreover, these results also open the possibility of finding similar associations in other combinations of viruses of agricultural or medical importance.

In contrast to satellite and virophages, that are obligate parasites that need to be coinfected with their counterpart viruses (*Palukaitis, Rezaian & García-Arenal, 2008*; *La Scola et al., 2008*), SeIV1 is capable of acting as a facultative phoretic parasite that can exploit the OBs produced by the alphabaculovirus to disperse and persist outside the host. In this sense, the alphabaculovirus OB can act as a vector for iflavirus transmission. As natural populations of Lepidoptera can harbor iflaviruses in the absence of baculovirus infection, it is clear that the association is not obligatory. However, the association is clearly advantageous for the transmission of the iflavirus. Indeed, other iflaviruses, namely *Ectropis obliqua virus* and *Perina nuda virus*, have been detected previously in mixed infections with an alphabaculovirus in their respective hosts (*Chen, Becnel & Valles, 2012*). Similarly, a small RNA virus was detected as a contaminant of *Autographa californica multiple nucleopolyhedrovirus* (AcMNPV) preparations (*Morris, Vail & Collier, 1981*). However, in none of these cases was any physical association of the viruses determined. It is tempting to speculate that because SeIV1 generally relies on vertical transmission to the offspring

of an infected insect, the iflavirus may benefit from its association with baculovirus OBs, because larvae that consume SeIV1-contaminated OBs are less likely to succumb to lethal polyhedrosis disease and may survive, reproduce and vertically transmit the iflavirus infection to their offspring.

We were not able to unambiguously localize iflavirus particles to the baculovirus OBs by transmission microscopy. However, structures resembling iflavirus particles by size and shape were observed in some preparations of the OBs that were positive for the presence of SeIV1. We believe that either whole iflavirus particles or viral RNA capable of infection is physically associated and may be occluded within the baculovirus OBs. Iflaviruses replicate in the cytoplasm whereas baculoviruses replicate in the nucleus. During infection baculovirus proteins are continuously imported from the cytoplasm into the nucleus (*Chen & Carstens, 2005*; *Au, Yu & Carstens, 2009*). SeIV1 particles or genomic RNA may be imported into the nucleus together with baculoviral proteins and be occluded within baculovirus OBs. Alternatively, the observed SeIV1-like particles could be the result of residual translocation of SeIV1 virions into the cell nucleus and most of the SeIV1 genomes detected by RT-qPCR of the OBs may be derived from naked RNA embedded in the OB matrix. We detected both positive and the negative strand of SeIV1 RNA, with a higher abundance of the former. This is consistent with the presence of naked RNA in the OBs, as the negative strand occurs only during iflavirus replication.

The discovery of novel viruses in all types of environments has increased markedly since the development of mass sequencing technologies (*Lecuit & Eloit, 2013*). Persistent infections are constantly being discovered in many insect species. Our studies suggest that an iflavirus may be able to employ the particles of another virus pathogen, a baculovirus, in order to increase virion persistence in the environment and as a vector to improve the likelihood of iflavirus transmission, decreasing pathogenicity of the baculovirus at the same time. Quantifying the impact of such insect virus associations on the ecology of both pathogens and host will require detailed field studies.

## ACKNOWLEDGEMENTS

We thank Anabel Millán-Leiva for technical assistance and Rafael Sanjuán for useful comments on the manuscript.

### Funding

This study received financial support from the Spanish Ministry for Science and Technology (AGL2011-30352-C02 and AGL2014-57752-C2). The funder had no role in study design, data collection and analysis, decision to publish, or preparation of the manuscript.

### Grant Disclosures

The following grant information was disclosed by the authors:
Spanish Ministry for Science and Technology: AGL2011-30352-C02, AGL2014-57752-C2.

## Competing Interests

The authors declare there are no competing interests.

## Author Contributions

- Agata K. Jakubowska conceived and designed the experiments, performed the experiments, analyzed the data, wrote the paper, prepared figures and/or tables, reviewed drafts of the paper.
- Rosa Murillo and Jan W.M. van Lent performed the experiments, analyzed the data.
- Arkaitz Carballo performed the experiments, analyzed the data, prepared figures and/or tables.
- Trevor Williams analyzed the data, wrote the paper, reviewed drafts of the paper.
- Primitivo Caballero analyzed the data.
- Salvador Herrero conceived and designed the experiments, analyzed the data, wrote the paper, prepared figures and/or tables, reviewed drafts of the paper.

## Data Availability

Raw data was provided as Supplemental Information.

## Supplemental Information

Supplemental information for this article can be found online at http://dx.doi.org/10.7717/peerj.1687#supplemental-information.

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
