# Peer review of "Iflavirus increases its infectivity and physical stability in association with baculovirus"

_PeerJ, doi:10.7717/peerj.1687_

## Round 0.1 · original submission · Minor Revisions

All three reviewers have provided constructive criticism that I think will help to strengthen this manuscript. One of the reviewers (Reviewer 2) in particular went very carefully through the manuscript and makes suggestions for controls that you should address, as well as other suggestions that, once addressed, will improve overall clarity of the manuscript. In addition to addressing the comments of the reviewers, here are a few minor comments of my own to address:

1. Lines 53-56: This is a very long sentence. I think it would read better if broken up into two sentences. My suggestion would be "..described in S. exigua. SelVI has a genome of.."
2. Line 197: I would add "a" as in "..resuspended in a small volume.."
3. Line 215: In the text, please comment on why this statistical test was chosen. Did the data not fit a normal distribution? Was it not known whether the data fit a normal distribution?
4. Line 223: Are words missing here? Should it say "..transferred to 24-well tissue culture plates?"

Reviewer 1 ·

Basic reporting

The manuscript was well written and is of interest to insect virologists since describes the possible association of two viruses of different families infecting the same host. The literature citation is appropriate and the figures are relevant for comprehension of the text.

Experimental design

The experiments were well designed and only one experiment lacks confirmation by a more accurate method. However, the authors acknowledge that and in my opinion the way the experiment was performed was sufficient to confirm what the authors claim.

Validity of the findings

The data presented are robust and confirm most of the conclusions.

Additional comments

The manuscript was well written and well performed and most conclusion are supported by the data presented. I have only a few comments

Line 110. Please remove the word 'and'

Line 158. Change the word 'above' to 'below'

Line 177. Please mention that the PCR product was visualized after electrophoresis in a 1% agarose gel.

Line 181. Change 'eletronic" to 'electron'

Lines 289-193. Please remove the text about Tomography, since no figure was added that showed this technique.

Lines 325-327. Please explain that these particles were purified from insect cadavers after several steps of ultracentrifugation at high speeds. This could have damaged the viral particles and influenced the infectivity of the virus preparation.

Reviewer 2 ·

Basic reporting

The article meets the journal required standards. It focuses on virus-virus interactions, an important subject in virology that needs more research.
To improve the manuscript, major additions are explained in Experimental Design and Validity of the Findings sections. A detailed list of changes is suggested in General Comments section.

Experimental design

A more detailed methodology is needed to explain how virus quantification was done. An extra subsection "Virus quantification" in Methods is recommended including how both, iflavirus and baculovirus, were quantified. RTqPCR description to estimate SeIV1 copies is not clear enough.

Validity of the findings

A control showing equivalents amounts of SeIV in both treatments at the beginning of the experiment is missing (Fig. 2A). Same problem for the SeMNPV comparison between 2 treatments (Fig. 2B); an initial measurement showing similar starting viral loads is needed. These are important controls to sustain the title of the paper, "increment in iflavirus infectivity".
Detection of negative strand of SeIV by RT-PCT needs more controls too.

Additional comments

Lines 4-5. Include the names of the viruses studied in the work, if not in the title at least in the abstract.
Lines 73-99. What’s the source of SeIV1 alone? Include it in this “Insects and viruses” section.
Line 82. Please clarify-rephrase. What do you mean by “spontaneous infections”? Is the source of the virus unknown?
Line 110. Correct typo to “baculovirus infection and the experiment”
Lines 94, 96, 97, 99, 105. How were the viruses quantified?
Line 156 (and 108, 251, 258). Please add more details on the RT-qPCR used for quantification of purified viruses. What standards and normalization procedures were used? Did you use a standard curve made of known amounts/copies of viral RNA?
Line 169. Include primers sequence and position in SeIV1 genome.
Line 158. Virus purification is described below this line.
Line 280. Osmiun tetroxide
Lines 152, 296-307. Briefly explain how the gene expression normalization works. Why VF treatment would have viral sequences (VT/VF is reported)? Are all the probes showing background basal signal in the VF material?
Line 313. The detection of negative strand of SeIV1 was done by semi-quantitative RT-PCR not by RTqPCR as indicated at the beginning of this paragraph. This negative strand detection is questionable (see next note), consider rephrase.
Lines 315, 653 and Fig 1A. Electrophoresis picture is pointless if not showing all the RT-PCR controls for negative strand detection, which is very difficult since among other controls, a control made of viral positive strand RNA needs to be shown free of amplification. Show the proper controls or remove the inset.
Lines 317-332, Figs. 2A, B. A control is missing showing equivalents amounts of SeIV in both treatments at the beginning of the experiment. No SeIV1 replication can be prove by just one time point, the 1.5 fold difference (Line 326- is NOT a “decrease” because you are measuring 2 different treatments; it’s then a 1.5 fold difference) could be due to differences in inocula amounts. Same problem for the SeMNPV comparison between 2 treatments (Fig. 2B); an initial measurement showing similar starting viral loads is needed.
Line 322. The methods section says 72 hpi, and here says 4 days, correct where needed.
Line 421. Some baculovirus transcription is shown in Fig. 1C, is not correct to say you didn’t detect any.
Line 645. Correct to “tiling”
Fig 1. Label Y axes as “log2 relative abundance”
Line 675 and Fig 4. Remove the uppercase labels A and B of the figure and the legend. Use labels A-H to describe the figure (in similar way to Fig. 3). Add arrows in figures G and H too.

Reviewer 3 ·

Basic reporting

- I suggest to add to Fig 1A that the numbers on the y-axis (relative abundance)are actually on a log2 scale, to make it clear that there is quite a big difference in rel abundance between pos and neg strand.
- In relation to this, I find the way that you present the small electrophoresis picture within Fig. 1A slightly odd; surely this gel is quite important evidence for co-occlusion of iflavirus in OBs, so why not make a separate panel for it? see comment below

Experimental design

no comments

Validity of the findings

- Your finding that the presence of iflavirus in association with SeMNPV seems to have a negative impact on baculovirus transcription is intriguing. Fig. 2D shows that OB production in larvae did not alter in the presence of iflavirus, but could it be that in OBs carrying iflavirus the number of ODVs/OB and/or the number of nucleocapsids/ODV was altered compared to iflavirus-free OBs? This could probably relatively easily be checked by comparing EM pictures as those shown in Fig. 3.
- I find the gel pic of Fig. 1A more convincing evidence for co-occlusion than the EM pics of Fig 3 (simply because the spots that should represent iflavirus are quite vague), so I suggest to give more attention to Fig 1A.

Additional comments

This paper describes the very interesting finding that viruses can 'hitchhike' with baculovirus particles, likely to increase their chance of persistence in the environment. Although there's clearly still a lot unknown about this association, this study provides an important first step. The manuscript is clearly written and easy to understand.

---

## Round 0.2 · Minor Revisions

Thank you for addressing the reviewers' comments and my own. There are a few small grammatical and formatting edits to be made, which are listed below:

1. Line 17 - it looks as though a word is missing here. I would suggest inserting "as" like so "..in the use of baculoviruses as biological insecticides."
2. Line 149 - should say "55-mers" (plural) not "55-mer" (singular)
3. Line 175 - it looks as though a word is missing here. I suggest inserting "fragment" or "amplicon" so that it reads "..to amplify a 97-bp fragment in.."
4. Line 177 - There is an extra space after 5'-
5. Lines 185-186. Please examine formatting here for presenting primer sequences in text. On the previous page, primer sequences are presented with a dash (-) between 5' or 3' and the sequence. Please pick a format and use it consistently throughout the manuscript.
6. Line 191 - I think this sentence would read more clearly if it said "..for 5 minutes.." rather than the comma between temperature and time.
7. Line 196 - I think there is a word missing here. I would insert "a" so that it reads "..loaded onto a 30-60 %.."
8. Line 204 - it says "..per gram of larva." Is it really larva (singular) or should it be larvae (plural)?
9. Lines 199-212. In this section, the first time x g is stated, it is in plain text. After that it is italicized throughout. Convert to plain text.
10. Line 206 - to make this sentence read more clearly, I would suggest inserting "then" so that it reads "..cheesecloth and then centrifuged.."
11. Line 221 - delete comma after "Both."
12. Line 237 - Insert "a" so that it reads "..transferred to a 24-well tissue culture plate."
13. Line 238 - Insert "a" so that it reads "..reared on a semisynthetic diet.."
14. Line 249 - For consistency, insert a dash (-) between 24 and well so that it reads "24-well tissue culture plate.." Also consider using the word "food" rather than "diet" here.
15. Line 250 - "weighed" not "weighted"
16. Line 265 - "were" not "was" as in "The amounts ..were estimated.."
17. Line 277 - I would consider a reword here, perhaps something like ".. and the samples were frozen until RNA purification."
18. Line 288 - "70% ethanol"
19. Line 293 - "0.4% agar"
20. Line 311 - rather than "..that indicated the presence.." consider saying ".. indicating the presence.."
21. Line 305 - it looks like there is extra space between S. and exigua
22. Lines 372-378 and in Figure 4 and associated legend - All the other figures have subpanels listed with capitalized letters rather than lowercase. This one should one should have capitalized letters as well.
23. Line 423 - "massively parallel.."
24. Make sure genus and species names are italicized throughout References list.

---

## Round 0.3 · accepted · Accept

Thank you for making that last round of edits.